# A Multitask Network for People Counting, Motion Recognition, and Localization Using Through-Wall Radar

**DOI:** 10.3390/s23198147

**Published:** 2023-09-28

**Authors:** Junyu Lin, Jun Hu, Zhiyuan Xie, Yulan Zhang, Guangjia Huang, Zengping Chen

**Affiliations:** School of Electronics and Communication Engineering, Sun Yat-sen University, Shenzhen 518107, China; linjy67@mail2.sysu.edu.cn (J.L.); xiezhy36@mail2.sysu.edu.cn (Z.X.); zhangylan23@mail2.sysu.edu.cn (Y.Z.); huanggj25@mail2.sysu.edu.cn (G.H.); chenzengp@mail.sysu.edu.cn (Z.C.)

**Keywords:** people counting, motion recognition, localization, Doppler signature, residual attention network, through-wall radar

## Abstract

Due to the outstanding penetrating detection performance of low-frequency electromagnetic waves, through-wall radar (TWR) has gained widespread applications in various fields, including public safety, counterterrorism operations, and disaster rescue. TWR is required to accomplish various tasks, such as people detection, people counting, and positioning in practical applications. However, most current research primarily focuses on one or two tasks. In this paper, we propose a multitask network that can simultaneously realize people counting, action recognition, and localization. We take the range–time–Doppler (RTD) spectra obtained from one-dimensional (1D) radar signals as datasets and convert the information related to the number, motion, and location of people into confidence matrices as labels. The convolutional layers and novel attention modules automatically extract deep features from the data and output the number, motion category, and localization results of people. We define the total loss function as the sum of individual task loss functions. Through the loss function, we transform the positioning problem into a multilabel classification problem, where a certain position in the distance confidence matrix represents a certain label. On the test set consisting of 10,032 samples from through-wall scenarios with a 24 cm thick brick wall, the accuracy of people counting can reach 96.94%, and the accuracy of motion recognition is 96.03%, with an average distance error of 0.12 m.

## 1. Introduction

In counterterrorism operations, disaster rescue, and other operations in urban environments, there is an urgent need for staff to detect and locate enemy personnel or victims hidden within buildings or behind obstacles. This is essential for ensuring the success of operations and the safety of staff. The urgent needs in both military and civilian domains have led to the development of various concealed target detection techniques based on sound signals [1], electromagnetic waves [2], infrared [3], etc. Among these techniques, electromagnetic wave detection has exhibited remarkable performance in terms of penetrating different types of walls, detection range, and detection accuracy. Additionally, electromagnetic waves do not carry visual information such as the appearance characteristics of the target, which can protect privacy. As a result, it has become a prominent research focus in the field of human perception behind walls.

Different electromagnetic waves achieve different levels of human action recognition performance. A novel approach [4] was proposed to detect and track human skeletons in real time using millimeter wave radar. The high bandwidth of radar ensures sufficient distance resolution. However, the high-frequency signals of radar limit its penetration capability. To balance distance resolution and penetration capability, we investigate ultra-wideband (UWB) low-frequency signals in this paper.

Human action recognition technology based on TWR is mainly divided into two parts: the image domain and the parameter domain, corresponding to imaging technology and feature parameter extraction, respectively. In the image domain, radar imaging algorithms include the backprojection algorithm [5], the time reversal algorithm [6,7], the compressed sensing algorithm [8,9,10], the inverse scattering algorithm [11,12], etc. Adib et al. [13] proposed a system, RF-Capture, that analyzes multiple reflection snapshots across time and combines their information to recover the limbs of the human body behind a wall. Based on the unprecedented resolution of body parts and positions, ${\mathrm{Xaver}}^{\mathrm{TM}}$ 1000 [14] can distinguish whether the object is an adult, a child, or an animal. It is also equipped with AI-based tracking of live target patterns. Combined with deep learning, radar-based research in the image domain has made significant progress in behavior recognition, usually using three-dimensional (3D) radar signal datasets. In [15], the authors proposed the first model, RF-Pose3D, for generating 3D human skeletons from radar signals. Yongkun Song et al. [16] were the first to propose a pose reconstruction method using low-frequency ultra-wideband MIMO radar as a detection sensor. Zhijie Zheng et al. [17,18] proposed RPSNet based on cross-modal learning to achieve human skeleton and shape recovery.

However, radar imaging requires a large amount of computing resources, and radar images require correspondingly large networks. This places high demands on radar transmission signals and hardware systems. In contrast, methods in the parameter domain generally only need single-channel radar, which is more suitable for real-time and portable applications. Therefore, the experimental datasets used in this study are based on 1D radar signals for computational efficiency. In the parameter domain, the traditional method is to recognize micro-Doppler features on the spectrogram through machine learning [19,20]. Most research using one-dimensional signals can only solve the task of motion recognition or people counting, which makes it difficult to meet the diverse needs of human perception behind walls. In contrast, research using three-dimensional radar can simultaneously obtain the number, position, and localization information of the targets. Therefore, this paper introduces a network that can simultaneously perform people counting, motion recognition, and static human localization, the remarkable performance of which is revealed in the results. The main contributions of our method can be summarized as follows:

1. We propose a multitask network based on a residual network and attention mechanism, which can achieve people counting, motion recognition, and localization by using RTD spectra generated from 1D radar signals.

2. We propose a novel multiscale spatial and channel attention module (MSCAM) to capture richer context information and adaptively enhance fine-grained features.

3. Based on the concept of multilabel classification, we propose a static human target localization method that can predict the positions of multiple people.

4. Extensive experiments validate the feasibility of radar-based joint people counting, action recognition, and static human localization, as well as the effectiveness of the multitask network in addressing these issues.

The remainder of this paper is organized as follows. Section 2 describes previous radar-based work related to people counting, motion recognition, and human localization. Section 3 presents the radar signal processing method. Section 4 provides detailed information about our multitask network and the method for static human localization. Section 5 presents the experimental details, results, and analysis. Finally, Section 6 summarizes the work reported in this paper.

## 2. Related Work

In recent years, radar-based person perception methods have received extensive attention, and numerous innovative works have been proposed in combination with deep learning. This section introduces these works in three categories: people counting, motion recognition, and human localization.

### 2.1. People Counting

Existing work based on 1D radar signals mainly employs artificial analysis or deep learning methods to accurately determine the number of people from radar echoes. In [21], Jeong Woo et al. proposed a people counting algorithm based on a probability density function of the amplitudes of the main pulses from the major clusters according to the number of people and distance. The algorithm can be operated in an indoor environment and an elevator with an average mean absolute error of less than one person. In [22], a novel convolutional neural network (CNN) was presented to classify the number of people, extracting features of the multiscale range–time maps transformed from the radar echoes. Choi et al. have focused on radar-based people counting research. In 2020, they proposed PCNet [23], a network that combines convolutional layers and long short-term memory to extract temporal and spatial features from processed radar signals. To address the overfitting issue of deep neural networks caused by insufficient data, they proposed RPCNet [24] in 2022, which relieves the training burden through a parameter-efficient design and achieves stabilized network convergence by leveraging unsupervised pretraining of a 1D network. In addition, they presented a novel approach that achieves robust people counting in both wide and small regions of interest [25]. The approach performs principal component analysis and appropriate normalization on information, combining modified CLEAN-based features in the range domain and energy-based features in the frequency domain.

### 2.2. Motion Recognition

In the parameter domain, most research leverages the powerful feature extraction capability of CNN to autonomously learn the behavioral features from radar signals, followed by motion classification.

In 2019, Kılıç et al. [26] directly fed raw 1D radar signals into a CNN to detect the presence of people behind walls and classify whether they were standing or sitting. In [27], RadarNet was proposed, which uses two 1D convolutional layers in place of short-time Fourier transform to obtain the representations of radar signals. Haiquan Chen et al. [28] proposed an end-to-end 1D CNN for human activity classification. The abovementioned methods directly use unprocessed 1D radar signals as datasets, preserving comprehensive feature information. Therefore, these methods exhibit high classification accuracy. In some studies, radar data were preprocessed to extract micro-Doppler features, distance features, etc. Yuan He et al. [29] employed radar micro-Doppler signatures and a multiscale residual attention network for simultaneous activity recognition and person recognition. In [30], different motion patterns were mapped into simple integer sequences through an autoencoder self-organized mapping network. Xiaqing Yang et al. [31] utilized distance profiles to realize real-time recognition of four actions through an autoencoder network and a gated recurrent unit network. Xiang Wang et al. [32] recognized action through a complex-valued CNN.

### 2.3. Human Localization

Recent research in human localization has primarily focused on two-dimensional (2D) and three-dimensional (3D) radar signals. To improve the accuracy of Doppler radar target localization, Xiaoyi Lin et al. [33] proposed a 2D human localization algorithm based on a high-order differential equation and Doppler processing. In [34], a curve-length method estimating the length of the I/Q signal trajectory was presented, aiming to enhance the sensitivity of phase-based human detection. In 2022, Jeongpyo et al. [35] introduced a parallel 1D CNN structure consisting of independent regression and classification models for 2D localization and pose recognition. However, the proposed structure only works in single-person scenarios. For multihuman scenarios, Wang Changlong et al. [36,37] successively proposed two through-wall 2D localization and behavior recognition systems based on a 3D CNN and 3D radar images. In [38], Shangyi Yang and Youngok Kim converted the problem of simultaneous position estimation and posture perception into an image classification problem through the use of 1D radar signals and a CNN.

The abovementioned approach usually treats people counting, motion recognition, and localization as different tasks. Each task requires a separate network for completion, which makes it challenging to meet the diverse needs of human perception behind walls. The network proposed in this paper achieves simultaneous prediction for people counting, motion recognition, and static human localization.

## 3. Signal Model

The wall-penetrating radar proposed in this paper adopts stepped-frequency continuous wave (SFCW). The formula of SFCW is expressed as follows:(1)$$s_{t}\left(t,k\right)=\mathrm{rect}\left(\frac{t-kT_{p}-T_{p}/2}{T_{p}}\right)\cdot \exp\left[j2\pi \left(f_{0}+k\Delta f\right)t\right]$$
where $T_{p}$ is the pulse width, $f_{0}$ is the initial frequency, Δf is a fixed frequency step, k=0,1,…,K−1, and *K* is the total number of frequency steps in a cycle.

The echo from a target can be expressed as:(2)$$s_{r}\left(t,k\right)=A_{r}\mathrm{rect}\left(\frac{t-kT_{p}-T_{p}/2-\tau_{r}}{T_{p}}\right)\cdot \exp\left[j2\pi \left(f_{0}+k\Delta f\right)\left(t-\tau_{r}\right)\right]$$
where $A_{r}$ is the amplitude factor and $\tau_{r}$ is the target echo delay.

In the fast time dimension, we sequentially perform frequency mixing and *K*-point inverse fast Fourier transform (IFFT) on $s_{r}\left(t,k\right)$ to obtain a one-dimensional distance vector of a single slow time period as follows:(3)$$h\left(l\right)=\frac{B_{r}}{K}\cdot \frac{\sin\left[\pi \left(l-K\Delta f\tau_{r}\right)\right]}{\sin\left[\frac{\pi }{K}\left(l-K\Delta f\tau_{r}\right)\right]}\cdot \varphi \left(\tau_{r}\right),l=0,1,\cdots K-1$$
(4)$$\varphi \left(\tau_{r}\right)=\exp\left(j\pi \frac{K-1}{K}l\right)\cdot \exp\left\{-j\pi \left[2f_{0}+\left(K-1\right)\Delta f\right]\tau_{r}\right\}$$
where $B_{r}$ is the amplitude factor and *l* is the radial distance gate. The peak position of |h(l)| corresponds to the target position. After processing multiple periods uniformly, we obtain 1D distance vectors and arrange them over time to form a range–time (RT) matrix. We apply moving average background cancellation for static clutter suppression in the RT matrix, yielding the normalized RT spectrum. For the through-wall dataset, we also need to perform wall compensation. Figure 1a shows the RT spectrum representing walking back and forth along the distance direction, based on which we can observe that the person turned back after moving away from the radar.

We perform *N*-point fast Fourier transform along the slow time direction in the RT matrix and obtain the range–Doppler (RD) matrix. After normalizing the RD matrix and using the zero-velocity channel zeroing method to suppress static clutters, we obtain the RD spectrum. Figure 1b illustrates the RD spectrum representing walking back and forth along the distance direction, based on which we can observe the person moving within a range of 5–10 m.

We truncate the RT spectrum and the RD spectrum along the range direction to obtain matrices with dimensions of 128 × 60 and 128 × 256, respectively. Specifically, the dimensions in the range direction, slow time direction, and frequency direction are 128, 60, and 256, respectively. Subsequently, we concatenate them along the range direction to obtain the RTD spectrum, the upper part of which is the RT spectrum. The y axis of the RTD spectrum represents the concatenation of the slow time direction and the frequency direction, so it has no units.

The datasets used in this study contain various activity types in non-through-wall and through-wall scenarios, such as various backgrounds, walking back and forth along the azimuth direction, walking back and forth along the distance direction, marking time, standing alone, and two people standing. Figure 2 shows the RTD spectra of six activity types. As shown in Figure 2a, there is no person in the scene. As shown in Figure 2b, the curve in the upper part of the image indicates that the person leaves and approaches the radar along the azimuth direction about 5 m away from the radar. In the lower part of the image, we can observe the motion range and Doppler frequency of the person, as well as multipath signals. As shown in Figure 2c, the green patches in the lower part of the image differ from those in Figure 2b. In Figure 2d, the green patches in the upper part of the image appear to be discontinuous, unlike those in Figure 2e. Furthermore, we can observe two persons in Figure 2f.

## 4. The Proposed Network

An overview of the network architecture for people counting, motion recognition, and static human localization is shown in Figure 3. The network is an end-to-end architecture composed of three modules: the backbone network, attention module, and confidence matrices for three tasks. As shown in Table 1, ResNet34 [39] is selected as the backbone network, with an MSCAM inserted after each ResNet stage. Initially, the network takes the RTD as input, passing it through a sequence of convolutional layers and MSCAM, thereby yielding enhanced feature representations. Subsequently, global average pooling is applied to integrate global high-level semantic features. Lastly, softmax layers are employed to transform the feature maps into confidence matrices for the tasks of people counting and motion recognition, with a sigmoid layer utilized to derive the predicted distance confidence matrix for static human localization.

### 4.1. Attention Module

In this section, we provide a detailed description of MSCAM, a lightweight module that effectively captures long-range dependencies in both spatial and channel dimensions.

As shown in Figure 3, MSCAM aggregates the multiscale spatial attention module (MSAM) and channel attention module (CAM) [40].

#### 4.1.1. Multiscale Spatial Attention Module

Spatial attention focuses on ‘where’, which is essential for scene understanding and complementary to channel attention. We introduce MSAM to model contextual relationships over local features, which can capture spatial dependencies between any two locations within the feature maps, thereby enhancing the representation capability of local features.

MSAM is shown in Figure 4a. We define the output of each ResNet stage as $A\in R^{C\times H\times W}$. We first feed A into convolutional layers to obtain four intermediate feature maps (B,C,D, and E, where $\left\{B,C\right\}\in R^{C/8\times H\times W}$ and $\left\{D,E\right\}\in R^{C/4\times H\times W}$). Afterwards, B,C,D, and E are reshaped to $R^{C_{1}\times N}$, where $C_{1}\in \left\{C/8,C/4\right\}$ and N=H×W. By transposing B and D, we perform matrix multiplication between B and C, as well as between D and E, resulting in two new feature maps (F and G). Subsequently, we add F and G together and apply a softmax layer to obtain the spatial attention map ($H\in R^{N\times N}$). Next, A is reshaped to $R^{C\times N}$ as the feature map (I), followed by matrix multiplication between I and H. The resulting matrix is reshaped and added to A, resulting in the final output ($S\in R^{C\times H\times W}$) as follows:(5)$$S=\lambda f_{R}\left(\mathrm{IH}\right)+A$$
where $f_{R}$ represents the reshape operation and λ learns a weight from zero. It can be observed that S at each position is a weighted sum of the features across all positions and the original features. Thus, it can selectively aggregate context based on the spatial attention map to obtain better representations of focused spatial features.

#### 4.1.2. Channel Attention Module

In contrast to MSAM, each channel map of high-level features is considered as a feature detector, focusing on ‘what’ is meaningful in the input data, and different feature detectors are interrelated. By exploiting the interdependencies among channel maps, we were able to improve the representation of focused features. Therefore, we use CAM to explicitly model the dependency among channels.

CAM [40] is shown in Figure 4b. Similarly to MSAM, we define the local feature as $A\in R^{C\times H\times W}$, but we directly reshape it as $B\in R^{N\times C}$, where N=H×W. Matrix multiplication is performed between C, which is the transpose of B, and B. After applying a softmax layer, we obtain the channel attention map (D). Then, we perform matrix multiplication between B and D and reshape the result to $R^{C\times H\times W}$, then add it to A to obtain the final output (S) as follows:(6)$$S=\mu f_{R}\left(\mathrm{DB}\right)+A$$
where μ also learns a weight from zero and S represents the weighted summation of the features from all channels and the original features, which highlights important semantic features.

### 4.2. Static Human Localization Method

In multilabel learning, each object is represented by a single instance and associated with a set of labels rather than a single label [41]. Motivated by multilabel learning, we introduce a method for static human localization based on distance confidence matrices in scenarios involving several persons, where a certain label corresponds to a certain position in the distance confidence matrix. A sample is associated with one or more labels, representing the positions of one or more persons in the sample. However, this method is not suitable for objects with large movement distances in a short period of time, as their high movement speed hinders accurate localization within the time intervals of accumulated pulses. Therefore, in this paper, we only locate people standing or marking time.

It is necessary to construct a distance confidence matrix based on the distance range of human targets. We intercept the RTD spectrum at $L_{0}$ points along the distance dimension and set the size of the distance confidence matrix $d_{tar}$ to 1×L, where each element has a value of 0. The maximum distance of an RTD spectrum with $L_{0}$ points in the distance dimension is $U_{0}$. The distance encoding formula is shown as (Equation 7), where *D* denotes the true distance of the target, $U_{d}$ is the true distance represented by a unit in the distance confidence matrix, and $U_{0}=U_{d}\times L$. The distance confidence matrices of the samples representing backgrounds are zero matrices, as are the samples that represent walking along the distance and azimuth directions.
(7)$$l=\left\lceil \frac{D}{U_{d}}\right\rceil$$
(8)$$d_{tar}\left[l-1\right]=1,l\in \left[0,L-1\right]$$

The network outputs the distance confidence matrix $d_{pre}$, which is the same size as $d_{tar}$. A higher score in $d_{pre}$ indicates a higher probability of the corresponding region containing a person. After setting the detection threshold θ, a cell in the confidence matrix after the sigmoid layer with a value exceeding θ indicates the presence of a target at the current location. To ensure that the network learns useful distance features, we incorporate θ into the loss function.

### 4.3. Loss Functions

The primary objective of the multitask network employed in this study is to accomplish three essential tasks within the given scene: people counting, action recognition, and static human localization. In order to achieve the goal, we need to not only implement parameter sharing through the network but also evaluate and optimize the model through the loss function during the training process. To address these tasks effectively, distinct loss functions were devised for each task, denoted as $loss_{Number}$, $loss_{Activity}$, and $loss_{Distance}$. $loss_{Number}$ and $loss_{Activity}$ are cross-entropy functions. $loss_{Distance}$ comprises two integral components, namely $loss_{Relative}$ and $loss_{True}$, as illustrated by (Equation 9). Notably, when computing $loss_{True}$, its value is contingent upon θ. If the confidence score ($d_{pre}$) subsequent to sigmoid layer processing surpasses θ, it is set to 1 before the loss function is calculated.. The overall loss function of the network is obtained by aggregating the aforementioned loss functions for the three tasks, as depicted in (Equation 12).
(9)$$loss_{Distance}=loss_{Relative}+loss_{True}$$
(10)$$\begin{array}{cc} loss_{Relative}=\; & -\frac{1}{L}\sum_{l=0}^{L-1}d_{tar}\left[l\right]\cdot \log\left({\left(1+\exp\left(-d_{pre}\left[l\right]\right)\right)}^{-1}\right)\\ \; & +\left(1-d_{tar}\left[l\right]\right)\cdot \log\left(\frac{\exp(-d_{pre}\left[l\right])}{\left(1+\exp\left(-d_{pre}\left[l\right]\right)\right)}\right) \end{array}$$
(11)$$loss_{True}=\sum_{l=0}^{L-1}\left|d_{tar}\left[l\right]-d_{pre}\left[l\right]\right|$$
(12)$$loss_{Total}=loss_{Number}+loss_{Activity}+loss_{Distance}$$

## 5. Experiment and Results

### 5.1. Implementation Details

#### 5.1.1. Datasets

The specific parameters of the ultra-wideband TWR used in the experiments are shown in Table 2, where PRF represents pulse repetition frequency. It can be observed from the table that the frequency band and bandwidth of the radar signal enable better penetration and distance resolution, and the pulse repetition period of the radar is also sufficient for indoor human target detection. The TWR contains 10 UWB patch antennas, with 2 serving as transmitting antennas and 8 serving as receiving antennae. The antenna array is a uniform linear array with two transmitting antennae at both ends of the array. We process raw radar echoes extracted from a single channel to obtain the datasets.

This study investigates two experimental scenarios: non-through-wall scenarios and through-wall scenarios with a wall thickness of 24 cm. The datasets comprise six distinct activity types performed by five people in these scenarios. The people stood or marked time in scenes within a Euclidean distance of 5.5 m from the radar origin. The people moved within the scene with a range of 3 m to the left and right of the radar origin and a depth of 10 m. Figure 5a shows the non-through-wall scenario for data training and testing. Figure 5b shows one of the non-through-wall scenarios used for data testing. Figure 5c,d show the through-wall scenarios for data testing.

Each sample consists of a sequence of 60 radar data frames. The training set comprises a total of 21,133 samples, while the test set, referred to as test set 1, consists of 10,032 samples from through-wall scenarios, and the other test set, denoted as test set 2, consists of 8416 samples from non-through-wall scenarios.

#### 5.1.2. Training Details

The network is implemented in PyTorch, using a stochastic gradient descent optimizer to update network parameters. For network training, we set the batch size to 32, the maximum learning rate to $1\times 10^{-3}$, and the momentum value to 0.9. Additionally, we use cosine annealing with the maximum number of iterations set to 20 and the detection threshold set to 0.01. All experiments are conducted on an RTX 3090 graphics card.

### 5.2. Performance of People Counting

The experimental results are shown in Figure 6. Based on test set 1, the detection rates of zero, one, and two persons reach 100%, 98.86%, and 87.34%, respectively. Meanwhile, based on test set 2, the detection rates for zero, one, and two persons are 99.92%, 99.68%, and 99.74%, respectively. As the distance between the targets and the radar increases, the amplitude of the target echoes gradually diminishes. In the through-wall scenarios of two people standing, the distant target may also experience partial occlusion by the foreground target and be occluded by the wall, resulting in further attenuation of their Doppler features and posing challenges for target recognition.

Based on test set 1, the results of the comparative experiments for each model are shown in Table 3. In [23,42], the RD spectrum was used as the dataset. It can be observed that in the through-wall scenarios, our model is far superior to the models proposed in [23,42] in terms of accuracy, precision, recall, and F1 score. This indicates that our model has strong generalization performance, and our data processing method reinforces the features related to the number of people. In addition, RPCNet [24] with restricted parameters fails to fit the datasets due to the sparse associations of its internal structures.

### 5.3. Performance of Motion Recognition

As depicted in Figure 7, the recognition accuracy of each motion in test set 2 exceeds 98%. However, the recognition accuracy of two people standing on test set 1 is relatively low, reaching only 86.03%. The reason for this is explained in the previous subsection. Table 4 presents the comparative experimental outcomes of various models on test set 1.

The accuracy of ID-1-D-CNN [28] is quite low when using an unprocessed 1D radar signal as input. This is because the raw radar signal contains various types of clutters, such as the waves transmitted from the surface in front of the wall and the waves reflected from the ceiling and the ground, which affect network learning and hinder cross-scene recognition. In comparison to the network based on an auto-encoder network and a gated recurrent unit [31], our model exhibits a notable improvement of over 8.9% in terms of accuracy, precision, recall, and F1 score. Moreover, our model outperforms the complex-value CNN [32] across all evaluated metrics, despite its considerably larger number of parameters compared to our model.

### 5.4. Performance of Static Human Localization

After correctly identifying the number of static individuals, we define the mean absolute localization error (MALE) of each person between the model predictions and the ground truth labels, which is calculated as follows:
(13)$$\mathrm{MALE}=\frac{\sum_{i=0}^{N-1}\left|D_{tar}\left[i\right]-D_{pre}\left[i\right]\right|}{N}$$
where $D_{tar}\left[i\right]$ and $D_{pre}\left[i\right]$ represent the true position and the predicted position of the person, respectively, and *N* is the total number of people in samples with correct results of static person count. In particular, $D_{pre}\left[i\right]$ is the median of the distance range corresponding to a specific region in the confidence score map.

Figure 8 presents the MALE of our method at different distances, indicating that our model has localization errors of 0.1204 m and 0.0997 m on test set 1 and test set 2, respectively. The localization error at 2–3 m based on test set 2 is relatively large, which is due to the multipath effect caused by the raised walls and ventilation pipes on the ceiling at 2–3 m in certain scenes.

Figure 9 displays the RTDs and the predicted localization results of the network proposed in this paper. The predicted results in the bottom row show people’s location coordinates and their corresponding true positions. The two columns on the left are the results on test set 1, showing the network’s capability of pinpointing multiple individuals. The two columns on the right are the results achieved on test set 2. The results illustrate that our method has good localization ability within the range of 5.5 m.

### 5.5. Effect of $loss_{True}$ on Model Performance

Setting appropriate loss functions is crucial for model optimization in this study. Consequently, we conduct experiments based on $loss_{True}$ to investigate the effectiveness of associating the loss function with θ.

Regarding the metrics, ${\mathrm{Acc}}_{\mathrm{no}}$, ${\mathrm{Acc}}_{\mathrm{motion}}$, and ${\mathrm{Acc}}_{\mathrm{loc}}$ represent the accuracy of people counting, motion recognition, and static person counting, respectively. The method of determining the values of $D_{pre}\left[i\right]$ leads to the accumulation of errors. As ${\mathrm{Acc}}_{\mathrm{loc}}$ increases, both $\sum_{i=0}^{N-1}\left|D_{tar}\left[i\right]-D_{pre}\left[i\right]\right|$ and *N* increase accordingly in (Equation 13).

Thus, predicting the changing trend of MALE becomes challenging. Considering that the primary goal of the human localization task is to locate as many individuals as possible rather than solely minimizing MALE, we prioritize using ${\mathrm{Acc}}_{\mathrm{loc}}$ as the primary evaluation metric while also taking MALE into account.

The effect of the proposed $loss_{True}$ is reported in Table 5 and Table 6. The experimental results demonstrate that with a constant detection threshold, $loss_{True}$ substantially enhances the overall performance of the model, especially in terms of accurately counting static individuals. However, this improvement is likely to be accompanied by an accumulation of average distance error.

### 5.6. Ablation Study

With the aim of analyzing the contribution of attention modules in the proposed network, we conduct various ablation experiments on datasets.

As depicted in Table 7 and Table 8, both CAM and MSAM contribute to improving the accuracy of various tasks, especially in ${\mathrm{Acc}}_{\mathrm{loc}}$. The aforementioned results illustrate that CAM autonomously learns the significance of each feature channel and assigns distinct weight coefficients to enhance feature representations. Furthermore, MSAM efficiently captures long-range contextual information to acquire discriminative features.

Based on Table 5, Table 6, Table 7 and Table 8, our model exhibits superior performance across multiple evaluation metrics in comparison to the baseline model, i.e., ResNet34 [39]. This indicates that MSCAM captures fine-grained features by enabling information interaction within spatial dimensions and across channels, thereby benefiting classification tasks and human localization.

## 6. Conclusions

In this paper, we propose an innovative multitask network for people counting, motion recognition, and static human localization. We employ RTD spectra acquired from 1D radar signals as datasets. We convert the positioning problem into a multilabel classification problem by creating distance confidence matrices grounded in the distance range of human targets. In these matrices, a particular position corresponds to a specific label, effectively encoding the spatial relationships. Convolutional layers and MSCAMs extract valuable spatial features and channel features from extensive RTD spectra, effectively establishing the correspondence between RTD spectra and their resultant outputs. In through-wall scenarios, our network achieves an accuracy of 96.94% for people counting and 96.03% for motion recognition. Furthermore, the average distance error is measured at 0.1204 m. In non-through-wall scenarios, the network achieves a people counting accuracy of 99.73% and a motion recognition accuracy of 99.37%. Additionally, the average distance error is measured at 0.0997 m.

In future work, we will focus on enhancing the precision of static human localization and develop new schemes for scenarios involving three or more individuals. Additionally, to better meet the requirements of practical deployment and applicability, we plan to design a lightweight network to further reduce the parameter count and floating-point operations.

## Figures and Tables

**Figure 1 sensors-23-08147-f001:**
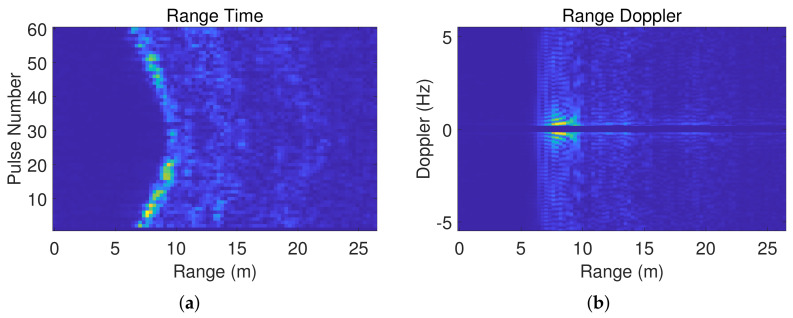
Walking back and forth along the distance direction. (**a**) RT spectrum; (**b**) RD spectrum.

**Figure 2 sensors-23-08147-f002:**
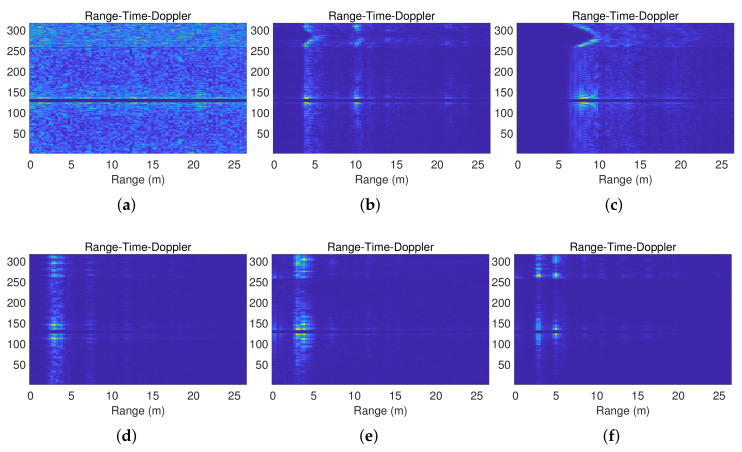
The RTD spectra. (**a**) Background; (**b**); walking back and forth along the azimuth direction; (**c**) walking back and forth along the distance direction; (**d**) marking time; (**e**) standing alone; (**f**) two people standing.

**Figure 3 sensors-23-08147-f003:**
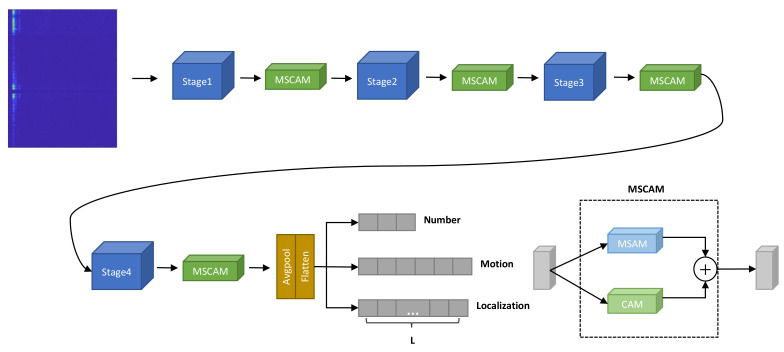
An overview of the network.

**Figure 4 sensors-23-08147-f004:**
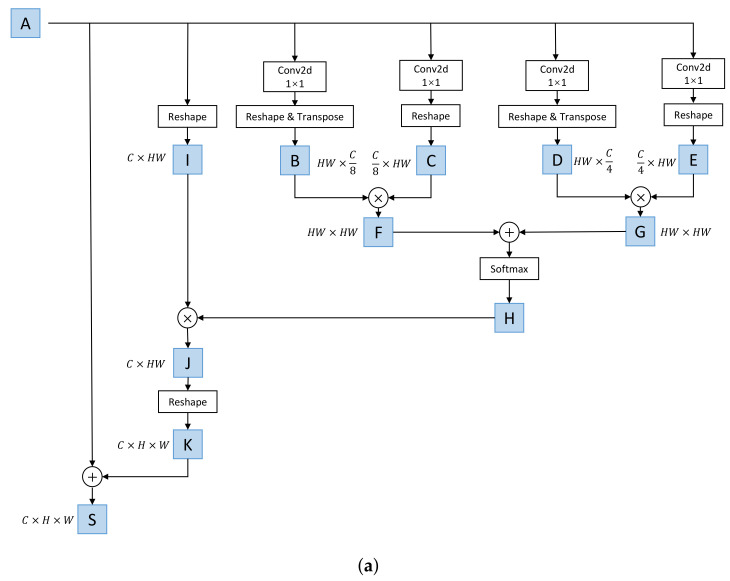
The architecture of MSCAM. (**a**) MSAM; (**b**) CAM [40].

**Figure 5 sensors-23-08147-f005:**
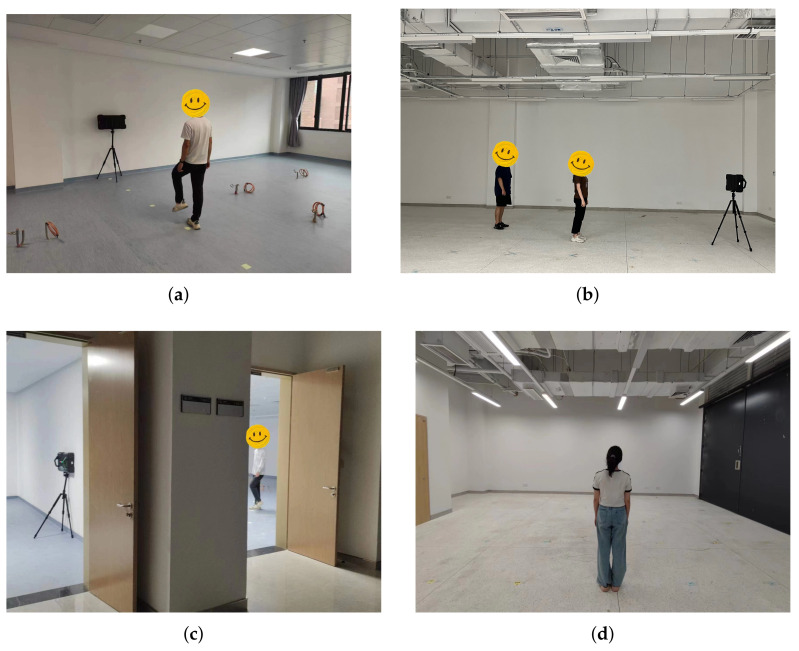
Experimental scenarios. (**a**) One of the non-through-wall scenarios; (**b**) one of the non-through-wall scenarios; (**c**) one of the through-wall scenarios; (**d**) one of the through-wall scenarios.

**Figure 6 sensors-23-08147-f006:**
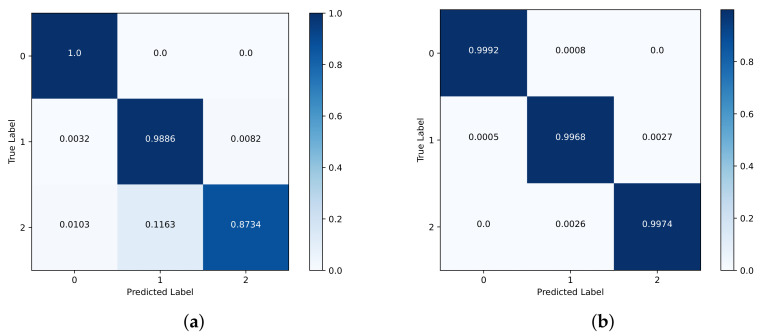
Confusion matrix for people counting. (**a**) Test set 1; (**b**) test set 2.

**Figure 7 sensors-23-08147-f007:**
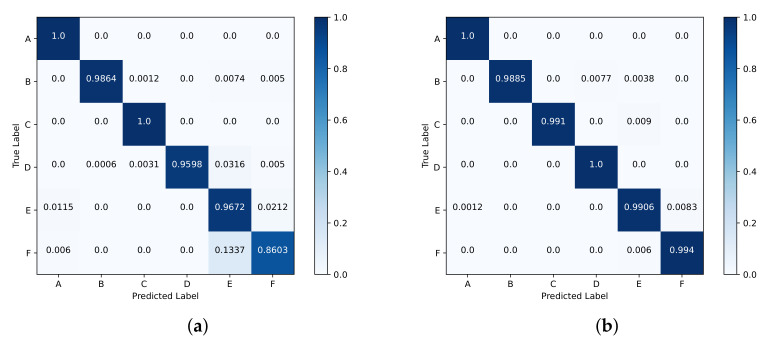
Confusion matrix for each motion. Classes codes represent six motions: A for background, B for marking time, C for walking along the range direction, D for walking along the azimuth direction, E for standing alone, and F for two people standing. (**a**) Test set 1; (**b**) test set 2.

**Figure 8 sensors-23-08147-f008:**
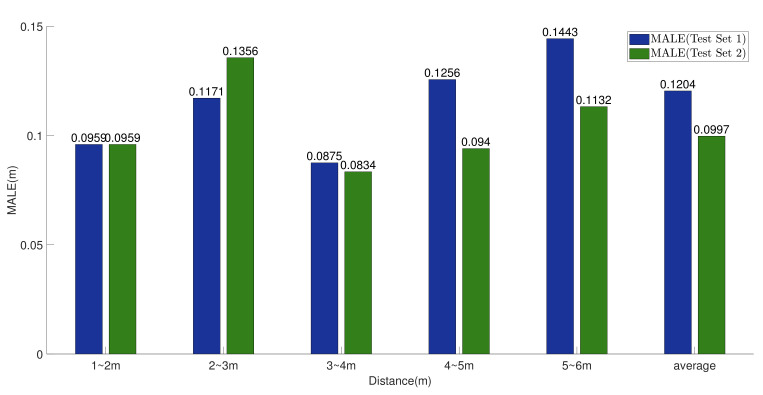
Positioning errors at different distances.

**Figure 9 sensors-23-08147-f009:**
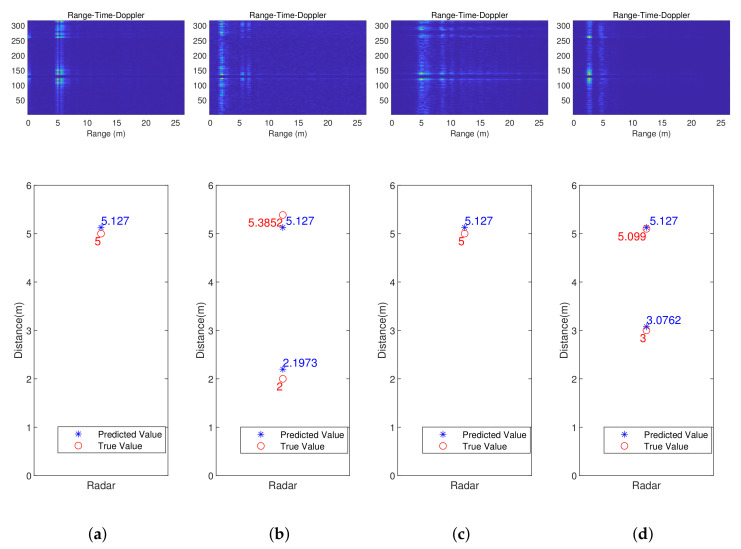
The visualized results of static human localization. (**a**) Standing alone (test set 1); (**b**) two people standing (test set 1); (**c**) marking time (test set 2); (**d**) Two people standing (test set 2).

**Table 1 sensors-23-08147-t001:** Stages in the network.

Layer Name	Operator
	conv, 7×7, 64, stride = 2
Stage 1	max pool, 3×3, stride = 2
	$\left[\begin{array}{cc} 3\times 3 & 64\\ 3\times 3 & 64 \end{array}\right]\times 3$
Stage 2	$\left[\begin{array}{cc} 3\times 3 & 128\\ 3\times 3 & 128 \end{array}\right]\times 4$
Stage 3	$\left[\begin{array}{cc} 3\times 3 & 256\\ 3\times 3 & 256 \end{array}\right]\times 6$
Stage 4	$\left[\begin{array}{cc} 3\times 3 & 512\\ 3\times 3 & 512 \end{array}\right]\times 3$

**Table 2 sensors-23-08147-t002:** The parameters of the radar.

Wave Form	Frequency Range	Step Frequency	Bandwidth	PRF
SFCW	1.900∼2.411 GHz	10 MHz	10 MHz	11

**Table 3 sensors-23-08147-t003:** Performance comparison of different models for people counting.

Model	Accuracy (%)	Precision (%)	Recall (%)	F1 Score (%)
ResNet14 [42]	91.42	92.42	85.45	87.10
PCNet [23]	93.43	93.15	89.70	90.85
RPCNet [24]	79.77	54.29	65.04	59.16
Ours (Test Set 1)	96.94	97.09	95.40	96.16
Ours (Test Set 2)	99.73	99.45	99.78	99.62

**Table 4 sensors-23-08147-t004:** Performance comparison of different models for motion recognition.

Model	Accuracy (%)	Precision (%)	Recall (%)	F1 Score (%)
ID-1-D-CNN [28]	20.07	16.55	19.32	15.82
AEN + GRU [31]	86.26	87.41	87.03	86.29
Complex-valued VGG16 [32]	92.17	92.56	92.23	92.34
Ours (Test Set 1)	96.03	96.40	96.23	96.18
Ours (Test Set 2)	99.37	99.39	99.40	99.39

**Table 5 sensors-23-08147-t005:** Effect of $loss_{True}$ on test set 1.

Loss	${\mathrm{Acc}}_{\mathrm{no}}$ (%)	${\mathrm{Acc}}_{\mathrm{motion}}$ (%)	${\mathrm{Acc}}_{\mathrm{loc}}$ (%)	MALE (m)
$loss_{Relative}$	95.95	94.47	72.96	0.0995
$loss_{Relative}+loss_{True}$	96.94	96.03	94.42	0.1204

**Table 6 sensors-23-08147-t006:** Effect of $loss_{True}$ on test set 2.

Loss	${\mathrm{Acc}}_{\mathrm{no}}$ (%)	${\mathrm{Acc}}_{\mathrm{motion}}$ (%)	${\mathrm{Acc}}_{\mathrm{loc}}$ (%)	MALE (m)
$loss_{Relative}$	99.04	98.22	82.92	0.0870
$loss_{Relative}+loss_{True}$	99.73	99.37	96.19	0.0997

**Table 7 sensors-23-08147-t007:** Ablation study of our method on test set 1.

Loss	${\mathrm{Acc}}_{\mathrm{no}}$ (%)	${\mathrm{Acc}}_{\mathrm{motion}}$ (%)	${\mathrm{Acc}}_{\mathrm{loc}}$ (%)	MALE (m)
ResNet34 [39]	95.69	94.27	87.54	0.1134
ResNet34 + CAM [40]	96.52	95.31	92.42	0.1194
ResNet34 + MSAM	96.33	95.23	91.40	0.1244

**Table 8 sensors-23-08147-t008:** Ablation study of our method on test set 2.

Loss	${\mathrm{Acc}}_{\mathrm{no}}$ (%)	${\mathrm{Acc}}_{\mathrm{motion}}$ (%)	${\mathrm{Acc}}_{\mathrm{loc}}$ (%)	MALE (m)
ResNet34 [39]	97.33	96.99	90.19	0.0964
ResNet34 + CAM [40]	99.05	98.50	95.42	0.0968
ResNet34 + MSAM	98.76	97.05	95.01	0.1055

## Data Availability

The data presented in this study are available on request from the corresponding author.

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
