# Peer review of "A Multitask Network for People Counting, Motion Recognition, and Localization Using Through-Wall Radar"

_sensors, 2023, doi:10.3390/s23198147_

Round 1

Reviewer 1 Report

The authors propose a system that simultaneously performs people counting, motion recognition and localization in a device-free manner.
They describe the system and evaluate with two datasets.
The results are promising, however, the main weakness of this paper is missing/incomplete description of the evaluation setup and procedure. Without this information, it is difficult to understand and ackowledge the contributions of the paper.

In the following are step by step remarks for improvement of the paper:

- In the abstract, the term positioning and localization is used interchangably. I recommend to decide for a single term.
- Please provide the material of the wall next the the wall thickness in line 25.
- I recommend to remove the university's name in line 85-86.
- In Section 2.1., 2.2. and 2.3., I am missing a brief description of you contributions and novelty after discussing related work.
- Which motions/activites were captured in the related work in Section 2.2, without this information it is difficult to evaluate the provided accuracy rates.
- In line 126 TWLBR and RTWLBR are not introduced.
- I recommend to provide sources for the equations (1)-(4).
- Figure 1a and Figure 1b. Increase the font size, and more important, zoom into the relevant signal ranges (from 0 - 20 m). Describe the figure more rigorously.
- Equation (4) $L$ is not introduced.
- Describe the magic numbers 128 x 60 and 128 x 256 in line 155.
- In line 157, I would hope to see that the datasets are provided in the supplementary material.
- Line 160 the term double standing is not clear to me. What do you mean with that?
- Figure 2, increase font size, provide y-axis description.
- Describe Figure 2 more, where do you observe subtle Doppler frequency features, I also recommend to zoom more into the relevant ranges from 0 - 30 m.
- MSCAM not introduced in line 173.
- Provide an example for high-level semantic features in line 175.
- Just by looking at Figure 3 i don't unterstand or acknowledge the claims in lines 182-184.
- In Section 4.1.1 Introduce $A,B,C,D,E,...$.
- Figure 3: Provide a description of the three stages. The block "Stage1,2,3" is very generic.
- Provide an example of the focused features in line 207.
- In line 227, I am missing a overall description of the test setup and description of the different experiments.
- In Section 5, I am missing details of the measurement hardware.
- Describe why you chose the provided values for the network training in line 279-280.
- In Section 5.3 the term behaviour is new for the reader. Use activity or motion instead?
- How did you handle the diameter of the persons in line 321?
- Figure 9 y-axis labels are missing, Why did you chose those explicit positions in your experiments?
- Section 5.6: is the term ablation correct?
- The list of abbreviations is incomplete.

Reviewer 2 Report

The existing imaging wall penetrating radar can quickly search for the movement of targets behind building wall obstacles, and can locate and observe from multiple angles or dimensions. It has functions such as target 3D imaging, positioning, and human target pose recognition, and has mature products. The wall penetrating radar designed by the authors requires prior knowledge and a dataset for training, and the designed method is limited in both application scope and scenario. Some drawbacks are as follows.

1.The author did not provide a clear introduction to the latest developments in wall penetrating radar.

2. The author did not address the engineering challenges faced by wall penetrating radar.

3.The neural network method designed by the author is not clearly written.

4.The rationality and universality of the loss function cannot be effectively proven.

5. How the author uses the network to achieve multitasking has not been clearly introduced.

6. The dataset is not authoritative enough.

7.When simulating, the simulation experiment is simple and not deep enough, and the credibility and superiority of the designed method and results cannot be guaranteed.

8.The universality of the designed method is not good enough, and its effectiveness cannot be guaranteed in another scenario.

9. The author did not conduct a thorough comparison with existing practical wall penetrating radars, and their superiority cannot be proven.

Some concepts are not clear enough and  the paper should be further polished.

Round 2

Reviewer 1 Report

The authors carefully read the reviewer's remarks and included it into their paper.

Reviewer 2 Report

The authors carefully revised the paper according to my suggestions, and now the paper can be accepted.

Now the paper can be read and understood easily.